# Adversarial Robustness Against the Union of Multiple Perturbation Models

## Abstract

Owing to the susceptibility of deep learning systems to adversarial attacks, there has been a great deal of work in developing (both empirically and certifiably) robust classifiers, but the vast majority has defended against single types of attacks. Recent work has looked at defending against multiple attacks, specifically on the MNIST dataset, yet this approach used a relatively complex architecture, claiming that standard adversarial training can not apply because it "overfits" to a particular norm. In this work, we show that it is indeed possible to adversarially train a robust model against a union of norm-bounded attacks, by using a natural generalization of the standard PGD-based procedure for adversarial training to multiple threat models. With this approach, we are able to train standard architectures which are robust against $\ell_\infty$, $\ell_2$, and $\ell_1$ attacks, outperforming past approaches on the MNIST dataset and providing the first CIFAR10 network trained to be simultaneously robust against $(\ell_\infty, \ell_2, \ell_1)$ threat models, which achieves adversarial accuracy of 46.1% against the union of $(\ell_\infty, \ell_2, \ell_1)$ perturbations with radius $\epsilon = (0.03, 0.5, 12)$.

## 1 Introduction

Machine learning algorithms have been shown to be susceptible to *adversarial examples* (Szegedy et al., 2014) through the existence of data points which can be adversarially perturbed to be misclassified, but are "close enough" to the original example to be imperceptible to the human eye. Methods to generate adversarial examples, or "attacks", typically rely on gradient information, and most commonly use variations of projected gradient descent (PGD) to maximize the loss within a small perturbation region, usually referred to as the adversary's threat model. Since then, a number of heuristic defenses have been proposed to defend against this phenomenon, e.g. distillation (Papernot et al., 2016) or more recently logit-pairing (Kannan et al., 2018). However, as time goes by, the original robustness claims of these defenses typically don't hold up to more advanced adversaries or more thorough attacks (Carlini & Wagner, 2017; Engstrom et al., 2018; Mosbach et al., 2018). One heuristic defense that seems to have survived (to this day) is to use *adversarial training* against a PGD adversary (Madry et al., 2018), which remains quite popular due to its simplicity and apparent empirical robustness. The method continues to perform well in empirical benchmarks even when compared to recent work in provable defenses, although it comes with no formal guarantees.

Some recent work, however, pointed out that adversarial training against $\ell_\infty$ perturbations "*overfits*" to the $\ell_\infty$ threat model, and used this as motivation to propose a more complicated architecture in order to achieve robustness to multiple perturbation types on the MNIST dataset (Schott et al., 2019).

In this work, we offer a alternative viewpoint: while adversarial training can overfit to the individual threat models, we show that it is indeed possible to use adversarial training to learn a model which is simultaneously robust against multiple types of $\ell_p$ norm bounded attacks (we consider $\ell_\infty$, $\ell_2$, and $\ell_1$ attacks, but the approach can apply to more general attacks). First, we show while simple generalizations of adversarial training to multiple threat models can achieve some degree of robustness against the union of these threat models, the performance is inconsistent and converges to suboptimal tradeoffs which may not actually minimize the robust objective. Second, we propose a slightly modified PGD-based algorithm called multi steepest descent (MSD) for adversarial training which more naturally incorporates the different perturbations within the PGD iterates, further improving the adversarial training approach by directly minimizing the robust optimization objective. Third, we show empirically that our approach improves upon past work by being applicable to standard

network architectures, easily scaling beyond the MNIST dataset, and outperforming past results on robustness against multiple perturbation types.

## 2 RELATED WORK

After their original introduction, one of the first widely-considered attacks against deep networks had been the Fast Gradient Sign Method (Goodfellow et al., 2015), which showed that a single, small step in the direction of the sign of the gradient could sometimes fool machine learning classifiers. While this worked to some degree, the Basic Iterative Method (Kurakin et al., 2017) (now typically referred to as the PGD attack) was significantly more successful at creating adversarial examples, and now lies at the core of many papers. Since then, a number of improvements and adaptations have been made to the base PGD algorithm to overcome heuristic defenses and create stronger adversaries. Adversarial attacks were thought to be safe under realistic transformations (Lu et al., 2017) until the attack was augmented to be robust to them (Athalye et al., 2018b). Adversarial examples generated using PGD on surrogate models can transfer to black box models (Papernot et al., 2017). Utilizing core optimization techniques such as momentum can greatly improve the attack success rate and transferability, and was the winner of the NIPS 2017 competition on adversarial examples (Dong et al., 2018). Uesato et al. (2018) showed that a number of ImageNet defenses were not as robust as originally thought, and Athalye et al. (2018a) defeated many of the heuristic defenses submitted to ICLR 2018 shortly after the reviewing cycle ended, all with stronger PGD variations.

Throughout this cycle of attack and defense, some defenses were uncovered that remain robust to this day. The aforementioned PGD attack, and the related defense known as adversarial training with a PGD adversary (which incorporates PGD-attacked examples into the training process) has so far remained empirically robust (Madry et al., 2018). Verification methods to certify robustness properties of networks were developed, utilizing techniques such as SMT solvers (Katz et al., 2017), SDP relaxations (Raghunathan et al., 2018b), and mixed-integer linear programming (Tjeng et al., 2019), the last of which has recently been successfully scaled to reasonably sized networks. Other work has folded verification into the training process to create provably robust networks (Wong & Kolter, 2018; Raghunathan et al., 2018a), some of which have also been scaled to even larger networks (Wong et al., 2018; Mirman et al., 2018; Gowal et al., 2018). Although some of these could potentially be extended to apply to multiple perturbations simultaneously, most of these works have focused primarily on defending against and verifying only a *single* type of adversarial perturbation at a time.

Last but most relevant to this work are adversarial defenses that attempt to be robust against multiple types of attacks simultaneously. Schott et al. (2019) used multiple variational autoencoders to construct a complex architecture for the MNIST dataset that is not as easily attacked by $\ell_\infty$, $\ell_2$, and $\ell_0$ adversaries. Importantly, Schott et al. (2019) compare to adversarial training with an $\ell_\infty$-bounded PGD adversary as described by Madry et al. (2018), claiming that the adversarial training defense overfits to the $\ell_\infty$ metric, and they do not consider other forms of adversarial training. Following this, a number of concurrent papers have since been released. While not studied as a defense, Kang et al. (2019) study the transferability of adversarial robustness between models trained against different threat models. Croce & Hein (2019) propose a provable adversarial defense against all $\ell_p$ norms for $p \geq 1$ using a regularization term. Finally, Tramèr & Boneh (2019) study the theoretical and empirical trade-offs of adversarial robustness in various settings when defending against multiple adversaries, however, they use a rotation and translation adversary instead of an $\ell_2$ adversary for CIFAR10.

**Contributions** In this work we demonstrate the effectiveness of adversarial training for learning models that are robust against a *union* of multiple perturbation models. First, we show that while simple aggregations of different adversarial attacks can achieve robustness against multiple perturbations models without resorting to complex architectures, the results are inconsistent across datasets and make suboptimal tradeoffs between the threat models. Second, we propose a modified PGD iteration that more naturally considers multiple perturbation models within the inner optimization loop of adversarial training. Third, we evaluate all approaches on the MNIST and CIFAR10 datasets, showing that our proposed generalizations of adversarial training can significantly outperform past approaches for the union of $\ell_\infty$, $\ell_2$, and $\ell_1$ attacks. Specifically, on MNIST, our model achieves 58.7% (individually 63.7%, 82.6%, 62.3%) adversarial accuracy against the union of all three attacks ($\ell_\infty, \ell_2, \ell_1$)

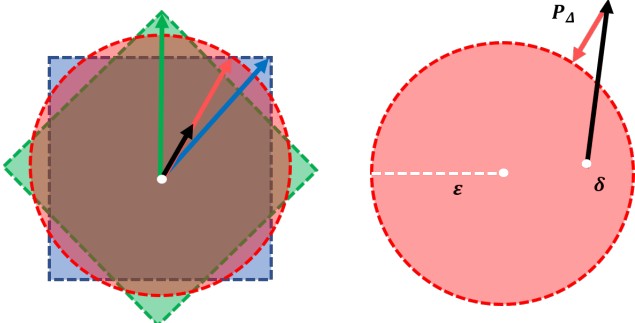

Figure 1: (left) A depiction of the steepest descent directions for $\ell_\infty$, $\ell_2$, and $\ell_1$ norms. The gradient is the black arrow, and the $\alpha$ radius step sizes and their corresponding steepest descent directions $\ell_\infty$, $\ell_2$, and $\ell_1$ are shown in blue, red, and green respectively. (right) An example of the projection back to an $\ell_2$ ball of radius $\epsilon$ after a steepest descent step from the starting perturbation $\delta$. The steepest descent step is the black arrow, and the corresponding projection back onto the $\ell_2$ ball is red arrow.

for $\epsilon = (0.3, 1.5, 12)$ respectively, substantially improving upon the multiple-perturbation-model robustness described in Schott et al. (2019) and also improving upon the simpler aggregations of multiple adversarial attacks. Unlike past work, we also train a CIFAR10 model, which achieves 46.1% (individually 47.6%, 64.3%, 53.4%) adversarial accuracy against the union of all three attacks $(\ell_\infty, \ell_2, \ell_1)$ for $\epsilon = (0.03, 0.5, 12)$. Finally, for completeness, we also draw relevant comparisons to concurrent work, and show that the relative advantage of our approach still holds.

## 3 Overview of adversarial training

Adversarial training is an approach to learn a classifier which minimizes the worst case loss within some perturbation region (the threat model). Specifically, for some network $f_\theta$ parameterized by $\theta$, loss function $\ell$, and training data $\{x_i, y_i\}_{i=1...n}$, the robust optimization problem of minimizing the worst case loss within $\ell_p$ norm-bounded perturbations with radius $\epsilon$ is

$$\min_\theta \sum_i \max_{\delta \in \Delta_{p,\epsilon}} \ell(f_\theta(x_i + \delta), y_i), \tag{1}$$

where $\Delta_{p,\epsilon} = \{\delta : \|\delta\|_p \leq \epsilon\}$ is the $\ell_p$ ball with radius $\epsilon$ centered around the origin. To simplify the notation, we will abbreviate $\ell(f_\theta(x + \delta), y) = \ell(x + \delta; \theta)$.

### 3.1 Solving the inner optimization problem

We first look at solving the inner maximization problem, namely

$$\max_{\delta \in \Delta_{p,\epsilon}} \ell(x + \delta; \theta). \tag{2}$$

This is the problem addressed by the "attackers" in the space of adversarial examples, hoping that the classifier can be tricked by the optimal perturbed image, $x + \delta^\star$. Typical solutions solve this problem by running a form of projected gradient descent, which iteratively takes steps in the gradient direction to increase the loss followed by a projection step back onto the feasible region, the $\ell_p$ ball. Since the gradients at the example points themselves (i.e., $\delta = 0$) are typically too small to make efficient progress, more commonly used is a variation called *projected steepest descent*.

**Steepest descent** For some norm $\|\cdot\|_p$ and step size $\alpha$, the direction of steepest descent on the loss function $\ell$ for a perturbation $\delta$ is

$$v_p(\delta) = \arg\max_{\|v\|_p \leq \alpha} v^T \nabla \ell(x + \delta; \theta). \tag{3}$$

Then, instead of taking gradient steps, steepest descent uses the following iteration

$$\delta^{(t+1)} = \delta^{(t)} + v_p(\delta^{(t)}). \tag{4}$$

In practice, the norm used in steepest descent is typically taken to be the same $\ell_p$ norm used to define the perturbation region $\Delta_{p,\epsilon}$. However, depending on the norm used, the direction of steepest descent can be quite different from the actual gradient (Figure 1). Note that a single steepest descent step with respect to the $\ell_\infty$ norm reduces to $v_\infty(x) = \alpha \cdot \text{sign}(\nabla \ell(x + \delta; \theta))$, better known in the adversarial examples literature as the Fast Gradient Sign Method (Goodfellow et al., 2015).

**Projections**    The second component of projected steepest descent for adversarial examples is to project iterates back onto the $\ell_p$ ball around $x$. Specifically, projected steepest descent performs the following iteration

$$\delta^{(t+1)} = \mathcal{P}_{\Delta_{p,\epsilon}} \left( \delta^{(t)} + v_p(\delta^{(t)}) \right) \tag{5}$$

where $\mathcal{P}_{\Delta_{p,\epsilon}}(\delta)$ is the standard projection operator that finds the perturbation $\delta' \in \Delta_{p,\epsilon}$ that is "closest" in Euclidean space to the input $\delta$, defined as

$$\mathcal{P}_{\Delta_{p,\epsilon}}(\delta) = \arg \min_{\delta' \in \Delta_{p,\epsilon}} \|\delta - \delta'\|_2^2. \tag{6}$$

Visually, a depiction of this procedure (steepest descent followed by a projection onto the perturbation region) for an $\ell_2$ adversary can be found in Figure 1. If we instead project the steepest descent directions with respect to the $\ell_\infty$ norm onto the $\ell_\infty$ ball of allowable perturbations, the projected steepest descent iteration reduces to

$$\begin{aligned} \delta^{(t+1)} &= P_{\Delta_{\infty,\epsilon}}(\delta^{(t)} + v_\infty(\delta^{(t)})) \\ &= \text{clip}_{[-\epsilon,\epsilon]} \left( \delta^{(t)} + \alpha \cdot \text{sign}(\nabla \ell(x + \delta^{(t)}; \theta)) \right) \end{aligned} \tag{7}$$

where $\text{clip}_{[-\epsilon,+\epsilon]}$ "clips" the input to lie within the range $[-\epsilon, \epsilon]$. This is exactly the Basic Iterative Method used in Kurakin et al. (2017), typically referred to in the literature as an $\ell_\infty$ PGD adversary.

## 3.2    Solving the outer optimization problem

We next look at how to solve the outer optimization problem, or the problem of learning the weights $\theta$ that minimize the loss of our classifier. While many approaches have been proposed in the literature, we will focus on a heuristic called adversarial training, which has generally worked well in practice.

**Adversarial training**    Although solving the min-max optimization problem may seem daunting, a classical result known as Danskin's theorem (Danskin, 1967) says that the gradient of a maximization problem is equal to the gradient of the objective evaluated at the optimum. For learning models that minimize the robust optimization problem from Equation (1), this means that

$$\nabla_\theta \left( \sum_i \max_{\delta \in \Delta_{p,\epsilon}} \ell(x_i + \delta; \theta) \right) = \sum_i \nabla_\theta \ell(x_i + \delta^*(x_i); \theta) \tag{8}$$

where $\delta^*(x_i) = \arg \max_{\delta \in \Delta_{p,\epsilon}} \ell(x_i + \delta; \theta)$. In other words, this means that in order to backpropagate through the robust optimization problem, we can solve the inner maximization and backpropagate through the solution. Adversarial training does this by empirically maximizing the inner problem with a PGD adversary. Note that since the inner problem is not solved exactly, Danskin's theorem does not strictly apply. However, in practice, adversarial training does seem to provide good empirical robustness, at least when evaluated against the $\ell_p$ threat model it was trained against.

## 4    Adversarial training for multiple perturbation models

We can now consider the core of this work, adversarial training procedures against multiple threat models. More formally, let $\mathcal{S}$ represent a set of threat models, such that $p \in \mathcal{S}$ corresponds to the $\ell_p$ perturbation model $\Delta_{p,\epsilon}$, and let $\Delta_{\mathcal{S}} = \bigcup_{p \in \mathcal{S}} \Delta_{p,\epsilon}$ be the union of all perturbation models in $\mathcal{S}$. Note that the $\epsilon$ chosen for each ball is *not* typically the same, but we still use the same notation $\epsilon$ for simplicity, since the context will always make clear which $\ell_p$-ball we are talking about. Then, the generalization of the robust optimization problem in Equation (1) to multiple perturbation models is

$$\min_\theta \sum_i \max_{\delta \in \Delta_{\mathcal{S}}} \ell(x_i + \delta; \theta). \tag{9}$$

The key difference is in the inner maximization, where the worst case adversarial loss is now taken over *multiple* $\ell_p$ perturbation models. In order to perform adversarial training, using the same motivational idea from Danskin's theorem, we can backpropagate through the inner maximization by first finding (empirically) the optimal perturbation,

$$\delta^* = \arg\max_{\delta \in \Delta_{\mathcal{S}}} \ell(x + \delta; \theta). \tag{10}$$

To find the optimal perturbation over the union of threat models, we begin by considering straightforward generalizations of standard adversarial training, which will use PGD to approximately solve the inner maximization over multiple adversaries.

## 4.1 SIMPLE COMBINATIONS OF MULTIPLE PERTURBATIONS

First, we study two simple approaches to generalizing adversarial training to multiple threat models. These methods can perform reasonably well in practice and are competitive with existing approaches without relying on complicated architectures. While these methods work to some degree, we later find empirically that these methods do not necessarily minimize the worst-case performance, and can converge to unexpected tradeoffs between multiple threat models.

**Worst-case perturbation**  One way to generalize adversarial training to multiple threat models is to use each threat model independently, and train on the adversarial perturbation that achieved the maximum loss. Specifically, for each adversary $p \in \mathcal{S}$, we solve the innermost maximization with an $\ell_p$ PGD adversary to get an approximate worst-case perturbation $\delta_p$,

$$\delta_p = \arg\max_{\delta \in \Delta_{p,\epsilon}} \ell(x + \delta; \theta), \tag{11}$$

and then approximate the maximum over all adversaries as

$$\delta^* \approx \arg\max_{\delta_p} \ell(x + \delta_p; \theta). \tag{12}$$

When $|\mathcal{S}| = 1$, then this reduces to standard adversarial training. Note that if each PGD adversary solved their subproblem from Equation (11) exactly, then this is exactly the optimal perturbation $\delta^\star$.

**PGD augmentation with all perturbations**  Another way to generalize adversarial training is to train on all the adversarial perturbations for all $p \in \mathcal{S}$ to form a larger adversarial dataset. Specifically, instead of solving the robust problem for multiple adversaries in Equation (9), we instead solve

$$\min_{\theta} \sum_i \sum_{p \in \mathcal{S}} \max_{\delta \in \Delta_{p,\epsilon}} \ell(x_i + \delta; \theta) \tag{13}$$

by using individual $\ell_p$ PGD adversaries to approximate the inner maximization for each threat model. Again, this reduces to standard adversarial training when $|\mathcal{S}| = 1$.

While these methods work reasonably well in practice (which is shown later in Section 5), both approaches solve the inner maximization problem independently for each adversary, so each individual PGD adversary is not taking advantage of the fact that the perturbation region is enlarged by other threat models. To take advantage of the full perturbation region, we propose a modification to standard adversarial training, which combines information from all considered threat models into a single PGD adversary that is potentially stronger than the combination of independent adversaries.

## 4.2 MULTI STEEPEST DESCENT

To create a PGD adversary with full knowledge of the perturbation region, we propose an algorithm that incorporates the different threat models within each step of projected steepest descent. Rather than generating adversarial examples for each threat model with separate PGD adversaries, the core idea is to create a single adversarial perturbation by simultaneously maximizing the worst case loss over all threat models at each projected steepest descent step. We call our method *multi steepest descent* (MSD), which can be summarized as the following iteration:

$$\delta_p^{(t+1)} = P_{\Delta_{p,\epsilon}}(\delta^{(t)} + v_p(\delta^{(t)})) \ \text{ for } \ p \in \mathcal{S}$$
$$\delta^{(t+1)} = \arg\max_{\delta_p^{(t+1)}} \ell(x + \delta_p^{(t+1)}) \tag{14}$$

---

**Algorithm 1** Multi steepest descent for learning classifiers that are simultaneously robust to $\ell_p$ attacks for $p \in \mathcal{S}$

---

**Input:** classifier $f_\theta$, data $x$, labels $y$
**Parameters:** $\epsilon_p, \alpha_p$ for $p \in \mathcal{S}$, maximum iterations $T$, loss function $\ell$
$\delta^{(0)} = 0$
**for** $t = 0 \ldots T - 1$ **do**
    **for** $p \in \mathcal{S}$ **do**
        $\delta_p^{(t+1)} = P_{\Delta_{p,\epsilon}}(\delta^{(t)} + v_p(\delta^{(t)}))$
    **end for**
    $\delta^{(t+1)} = \arg\max_{\delta_p^{(t+1)}} \ell(f_\theta(x + \delta_p^{(t+1)}), y)$
**end for**
return $\delta^{(T)}$

---

The key difference here is that at each iteration of MSD, we choose a projected steepest descent direction that maximizes the loss over all attack models $p \in \mathcal{S}$, whereas standard adversarial training and the simpler approaches use comparatively myopic PGD subroutines that only use one threat model at a time. The full algorithm is in Algorithm 1, and can be used as a drop in replacement for standard PGD adversaries to learn robust classifiers with adversarial training. We direct the reader to Appendix A for a complete description of steepest descent directions and projection operators for $\ell_\infty$, $\ell_2$, and $\ell_1$ norms[1].

## 5 RESULTS

In this section, we present experimental results on using generalizations of adversarial training to achieve simultaneous robustness to $\ell_\infty$, $\ell_2$, and $\ell_1$ perturbations on the MNIST and CIFAR10 datasets. Our primary goal is to show that adversarial training can in fact be adapted to a union of perturbation models using standard architectures to achieve competitive results, without the pitfalls described by Schott et al. (2019). Our results improve upon the state-of-the-art in three key ways. First, we can use simpler, standard architectures for image classifiers, without relying on complex architectures or input binarization. Second, our method is able to learn a single MNIST model which is simultaneously robust to all three threat models, whereas previous work was only robust against two at a time. Finally, our method is easily scalable to datasets beyond MNIST, providing the first CIFAR10 model trained to be simultaneously robust against $\ell_\infty$, $\ell_2$, and $\ell_1$ adversaries.

We trained models using both the simple generalizations of adversarial training to multiple adversaries and also using MSD. Since the analysis by synthesis model is not scalable to CIFAR10, we additionally trained CIFAR10 models against individual PGD adversaries to measure the changes and tradeoffs in universal robustness. We evaluated these models with a broad suite of both gradient and non-gradient based attacks using Foolbox[2] (the same attacks used by Schott et al. (2019)), and also incorporated all the PGD-based adversaries discussed in this paper. All aggregate statistics that combine multiple attacks compute the worst case error rate over all attacks for *each* example.

Summaries of these results at specific thresholds can be found in Tables 1 and 2, where B-ABS and ABS refer to binarized and non-binarized versions of the analysis by synthesis models from Schott et al. (2019), $P_p$ refers to a model trained against a PGD adversary with respect to the $p$-norm, Worst-PGD and PGD-Aug refer to models trained using the worst-case and data augmentation generalizations of adversarial training, and MSD refers to models trained using multi steepest descent. Full tables containing the complete breakdown of these numbers over all individual attacks used in the evaluation are in Appendix C. We report the results against individual attacks and threat models for completeness, however note that the goal of all these algorithms is to minimize the robust optimization objective from Equation (9). While there may be different implicit tradeoffs between individual threat

---

[1]The pure $\ell_1$ steepest descent step is inefficient since it only updates one coordinate at a time. It can be improved by taking steps on multiple coordinates, similar to that used in Tramèr & Boneh (2019), and is also explained in Appendix A.
[2]https://github.com/bethgelab/foolbox (Rauber et al., 2017)

Table 1: Summary of adversarial accuracy results for MNIST (higher is better)

| | $P_\infty$ | $P_2$ | $P_1$ | B-ABS[4] | ABS[4] | Worst PGD | PGD Aug | MSD |
|---|---|---|---|---|---|---|---|---|
| Clean Accuracy | 99.1% | 99.4% | 98.9% | 99% | 99% | 98.9% | 99.1% | 98.2% |
| $\ell_\infty$ attacks ($\epsilon = 0.3$) | 90.3% | 0.4% | 0.0% | 77% | 8% | 68.4% | 83.7% | 63.7% |
| $\ell_2$ attacks ($\epsilon = 1.5$) | 45.3% | 87.0% | 70.3% | 39% | 80% | 82.1% | 75.0% | 82.6% |
| $\ell_1$ attacks ($\epsilon = 12$) | 1.4% | 43.4% | 71.8% | 82% | 78% | 54.6% | 15.6% | 62.3% |
| All Attacks | 1.4% | 0.4% | 0.0% | 39% | 8% | 53.7% | 15.6% | **58.7%** |

models, in the end, the most meaningful metric for measuring the effective performance is the robust optimization objective, or the performance against the union of *all* attacks.

## 5.1 EXPERIMENTAL SETUP

**Architectures and hyperparameters** For MNIST, we use a four layer convolutional network with two convolutional layers consisting of 32 and 64 $5 \times 5$ filters and 2 units of padding, followed by a fully connected layer with 1024 hidden units, where both convolutional layers are followed by $2 \times 2$ Max Pooling layers and ReLU activations (this is the same architecture used by Madry et al. (2018)). This is in contrast to past work on MNIST, which relied on per-class variational autoencoders to achieve robustness against multiple threat models (Schott et al., 2019), which was also not easily scalable to larger datasets. Since our methods have the same complexity as standard adversarial training, they also easily apply to standard CIFAR10 architectures, and in this paper we use the well known pre-activation version of the ResNet18 architecture consisting of nine residual units with two convolutional layers each (He et al., 2016).

A complete description of the hyperparameters used is in Appendix B, with hyperparameters for PGD adversaries in Appendix B.1, and hyperparameters for adversarial training in Appendix B.2. All reported $\epsilon$ are for images scaled to be between the range $[0, 1]$. All experiments can be run on modern GPU hardware (e.g. a single 1080ti).

**Attacks used for evaluation** To evaluate the model, we incorporate the attacks from Schott et al. (2019) as well as our PGD based adversaries using projected steepest descent, however we provide a short description here. Note that we exclude attacks based on gradient estimation, since the gradient for the standard architectures used here are readily available.

For $\ell_\infty$ attacks, although we find the $\ell_\infty$ PGD adversary to be quite effective, for completeness, we additionally use the Foolbox implementations of Fast Gradient Sign Method (Goodfellow et al., 2015), PGD adversary (Madry et al., 2018), and the Momentum Iterative Method (Dong et al., 2018).

For $\ell_2$ attacks, in addition to the $\ell_2$ PGD adversary, we use the Foolbox implementations of the same PGD adversary, the Gaussian noise attack (Rauber et al., 2017), the boundary attack (Brendel et al., 2017), DeepFool (Moosavi-Dezfooli et al., 2016), the pointwise attack (Schott et al., 2019), DDN based attack (Rony et al., 2018), and C&W attack (Carlini & Wagner, 2017).

For $\ell_1$ attacks, we use both the $\ell_1$ PGD adversary as well as additional Foolbox implementations of $\ell_0$ attacks at the same radius, namely the salt & pepper attack (Rauber et al., 2017) and the pointwise attack (Schott et al., 2019). Note that an $\ell_1$ adversary with radius $\epsilon$ is strictly stronger than an $\ell_0$ adversary with the same radius, and so we choose to explicitly defend against $\ell_1$ perturbations instead of the $\ell_0$ perturbations considered by Schott et al. (2019).

We make **10 random restarts** for each of the evaluation results mentioned hereon for both MNIST and CIFAR10 [3]. We encourage future work in this area to incorporate the same, since the success of all attacks, specially decision based or gradient free ones, is observed to increase significantly over restarts.

---

[3] All attacks were run on a subset of the first 1000 test examples with 10 random restarts, with the exception of Boundary Attack, which by default makes 25 trials per iteration and DDN based Attack which does not benefit from the same owing to a deterministic initialization of $\delta$.

[4] Results are from Schott et al. (2019), which used an $\ell_0$ threat model of the same radius and evaluated against $\ell_0$ attacks. So the reported number here is an upper bound on the $\ell_1$ adversarial accuracy. Further, they evaluate

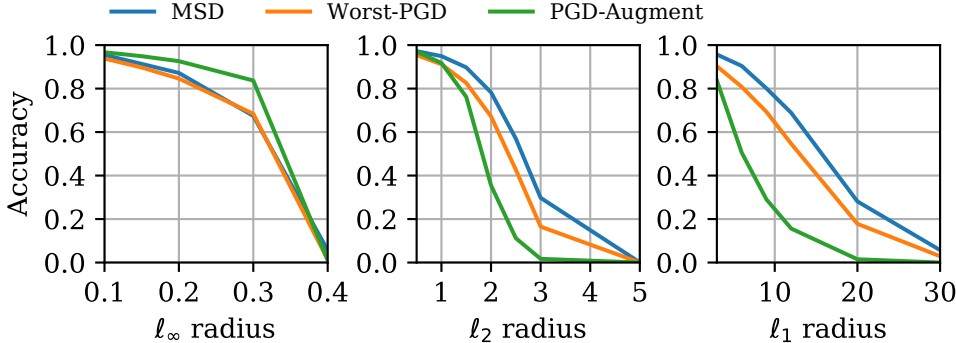

Figure 2: Robustness curves showing the adversarial accuracy for the MNIST model trained with MSD, PGD-Aug, Worst-PGD against $\ell_\infty$ (left), $\ell_2$ (middle), and $\ell_1$ (right) threat models over a range of epsilon.

Table 2: Summary of adversarial accuracy results for CIFAR10 (higher is better)

|  | $P_\infty$ | $P_2$ | $P_1$ | Worst-PGD | PGD-Aug | MSD |
|---|---|---|---|---|---|---|
| Clean accuracy | 83.3% | 90.2% | 73.3% | 81.0% | 84.6% | 81.7% |
| $\ell_\infty$ attacks ($\epsilon = 0.03$) | 50.7% | 28.3% | 0.2% | 44.9% | 42.5% | 47.6% |
| $\ell_2$ attacks ($\epsilon = 0.5$) | 57.3% | 61.6% | 0.0% | 61.7% | 65.0% | 64.3% |
| $\ell_1$ attacks ($\epsilon = 12$) | 16.0% | 46.6% | 7.9% | 39.4% | 54.0% | 53.4% |
| All attacks | 15.6% | 27.5% | 0.0% | 34.9% | 40.6% | **46.1%** |

## 5.2 MNIST

We first present results on the MNIST dataset, which are summarized in Table 1 (a more detailed breakdown over each individual attack is in Appendix C.1). While considered an "easy" dataset, we note that the previous state-of-the-art result for multiple threat models on MNIST (and our primary comparison) is only able to defend against two out of three threat models at a time (Schott et al., 2019) using comparatively complex variational autoencoder architectures. The model trained with MSD achieves the best performance against all attacks, achieving an error rate of 58.7% (individually 63.7%, 82.6%, and 62.3)% against the union of ($\ell_\infty$, $\ell_2$, and $\ell_1$) perturbations with radius $\epsilon = (0.3, 1.5, 12)$. Complete robustness curves over a range of epsilons over each threat model can be found in Figure 2. A comparison of our results with concurrent work (Tramèr & Boneh, 2019) can be found in Appendix D.

## 5.3 CIFAR10

Next, we present results on the CIFAR10 dataset, which are summarized in Table 2 (a more detailed breakdown over each individual attack is in Appendix C.2). Our MSD approach reaches the best performance against the union of attacks, and achieves $46.1\%$ (individually $47.6\%, 64.3\%, 53.4\%$) adversarial accuracy against the union of ($\ell_\infty$, $\ell_2$, $\ell_1$) perturbations of size $\epsilon = (0.03, 0.5, 12)$. Interestingly, note that the $P_1$ model trained against an $\ell_1$ PGD adversary is not very robust when evaluated against other attacks, even though it can defend reasonably well against the $\ell_1$ PGD attack in isolation (Table 4 in Appendix C.2). Complete robustness curves over a range of epsilons over each threat model can be found in Figure 3. A comparison of our results with concurrent work (Tramèr & Boneh, 2019) can be found in Appendix D. While adversarial defenses are generally not intended to defend against attacks outside of the threat model, we show some experiments exploring this aspect in Appendix E.

---

their model without restarts and the adversarial accuracy against all attacks is an upper bound based on the reported accuracies for the individual threat models. Finally, all ABS results were computed using numerical gradient estimation, since gradients are not readily available.

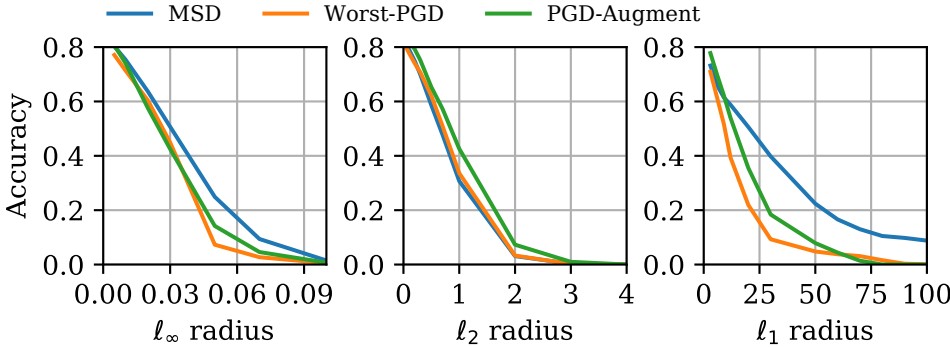

Figure 3: Robustness curves showing the adversarial accuracy for the CIFAR10 model trained with MSD, PGD-Aug, Worst-PGD against $\ell_\infty$ (left), $\ell_2$ (middle), and $\ell_1$ (right) threat models over a range of epsilon.

**On tradeoffs and variability of the simpler defenses** One major drawback to the simpler methods for generalizing adversarial training to multiple threat models is their variability and unclear tradeoffs over different settings. For example, on MNIST we see that the data augmentation approach fails to reduce the robust optimization objective: the $\ell_\infty$ threat model dominates the training process and we get a suboptimal tradeoff between threat models which isn't robust to the union. Similarly, on CIFAR10 we see that the worst-case approach for adversarial training also converges to a model which has suboptimal robust performance against the union of threat models. This highlights the inconsistency of the simpler generalizations of adversarial training: depending on the dataset and the threat models, they may not ultimately minimize the robust optimization objective from Equation (9), and the tradeoffs may vary significantly with the problem setting. On the other hand, in both problem settings, we find MSD is consistent at finding a more optimal tradeoff which minimizes the worst-case loss in the union of the threat models. As a result, rather than using one of the simpler methods and convergence to a potentially unclear tradeoff between threat models, we recommend using MSD which directly minimizes the worst case performance among the specified threat models.

## 6 CONCLUSION

In this paper, we showed that adversarial training can be quite effective when training against a union of multiple perturbation models. We compare two simple generalizations of adversarial training and an improved adversarial training procedure, multi steepest descent, which incorporates the different perturbation models directly into the direction of steepest descent. MSD based adversarial training procedure is able to outperform past approaches, demonstrating that adversarial training can in fact learn networks that are robust to multiple perturbation models simultaneously (as long as they are included in the threat model) while being scalable beyond MNIST and using standard architectures.

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

---

**Algorithm 2** Projection of some perturbation $\delta \in \mathbb{R}^n$ onto the $\ell_1$ ball with radius $\epsilon$. We use $|\cdot|$ to denote element-wise absolute value.

---

**Input:** perturbation $\delta$, radius $\epsilon$
Sort $|\delta|$ into $\gamma : \gamma_1 \geq \gamma_2 \geq \cdots \geq \gamma_n$
$\rho := \max \left\{ j \in [n] : \gamma_j - \frac{1}{j} \left( \sum_{r=1}^{j} \gamma_r - \epsilon \right) > 0 \right\}$
$\eta := \frac{1}{\rho} \left( \sum_{i=1}^{\rho} \gamma_i - \epsilon \right)$
$z_i := \text{sign}(\delta_i) \max \{ \gamma_i - \eta, 0 \}$ for $i = 1 \ldots n$
**return** $z$

---

## A    STEEPEST DESCENT AND PROJECTIONS FOR $\ell_\infty$, $\ell_2$, AND $\ell_1$ ADVERSARIES

In this section, we show what the steepest descent and projection steps are for $\ell_p$ adversaries for $p \in \{\infty, 2, 1\}$; these are standard results, but included for a complete description of the algorithms. Note that this differs slightly from the adversaries considered in Schott et al. (2019): while they used an $\ell_0$ adversary, we opted to use an $\ell_1$ adversary with the same radius. The $\ell_0$ ball with radius $\epsilon$ is contained within an $\ell_1$ ball with the same radius, so achieving robustness against an $\ell_1$ adversary is strictly more difficult.

$\ell_\infty$ **space**    The direction of steepest descent with respect to the $\ell_\infty$ norm is

$$v_\infty(\delta) = \alpha \cdot \text{sign}(\nabla l(x + \delta; \theta)) \tag{15}$$

and the projection operator onto $\Delta_{\infty,\epsilon}$ is

$$\mathcal{P}_{\Delta_{\infty,\epsilon}}(\delta) = \text{clip}_{[-\epsilon,\epsilon]}(\delta) \tag{16}$$

$\ell_2$ **space**    The direction of steepest descent with respect to the $\ell_2$ norm is

$$v_2(\delta) = \alpha \cdot \frac{\nabla \ell(x + \delta; \theta)}{\|\nabla \ell(x + \delta; \theta)\|_2} \tag{17}$$

and the projection operator onto the $\ell_2$ ball around $x$ is

$$\mathcal{P}_{\Delta_{2,\epsilon}}(\delta) = \epsilon \cdot \frac{\delta}{\max\{\epsilon, \|\delta\|_2\}} \tag{18}$$

$\ell_1$ **space**    The direction of steepest descent with respect to the $\ell_1$ norm is

$$v_1(\delta) = \alpha \cdot \text{sign} \left( \frac{\partial \ell(x + \delta; \theta)}{\partial \delta_{i^\star}} \right) \cdot e_{i^\star} \tag{19}$$

where

$$i^\star = \arg \max_i |\nabla l(x + \delta; \theta)_i| \tag{20}$$

and $e_{i^*}$ is a unit vector with a one in position $i^*$. Finally, the projection operator onto the $\ell_1$ ball,

$$\mathcal{P}_{\Delta_{1,\epsilon}}(\delta) = \arg \min_{\delta' : \|\delta'\|_1 \leq \epsilon} \|\delta - \delta'\|_2^2, \tag{21}$$

can be solved with Algorithm 2, and we refer the reader to Duchi et al. (2008) for its derivation.

### A.1    ENHANCED $\ell_1$ STEEPEST DESCENT STEP

Note that the steepest descent step for $\ell_1$ only updates a single coordinate per step. This can be quite inefficient, as pointed out by Tramèr & Boneh (2019). To tackle this issue, and also empirically improve the attack success rate, Tramèr & Boneh (2019) instead select the top $k$ coordinates according to Equation 20 to update. In this work, we adopt a similar but slightly modified scheme: we randomly sample $k$ to be some integer within some range $[k_1, k_2]$, and update each coordinate with step size $\alpha' = \alpha/k$. We find that the randomness induced by varying the number of coordinates aids in avoiding the gradient masking problem observed by Tramèr & Boneh (2019).

## A.2 RESTRICTING THE STEEPEST DESCENT COORDINATE

The steepest descent direction for both the $\ell_0$ and $\ell_1$ norm end up selecting a single coordinate direction to move the perturbation. However, if the perturbation is already at the boundary of pixel space (for MNIST, this is the range [0,1] for each pixel), then it's possible for the PGD adversary to get stuck in a loop trying to use the same descent direction to escape pixel space. To avoid this, we only allow the steepest descent directions for these two attacks to choose coordinates that keep the image in the range of real pixels.

## B EXPERIMENTAL DETAILS

### B.1 HYPERPARAMETERS FOR PGD ADVERSARIES

In this section, we describe the parameters used for all PGD adversaries in this paper.

**MNIST**   The $\ell_\infty$ adversary used a step size $\alpha = 0.01$ within a radius of $\epsilon = 0.3$ for 50 iterations.

The $\ell_2$ adversary used a step size $\alpha = 0.1$ within a radius of $\epsilon = 1.5$ for 100 iterations.

The $\ell_1$ adversary used a step size of $\alpha = 0.05$ within a radius of $\epsilon = 12$ for 50 iterations. By default the attack is run with two restarts, once starting with $\delta = 0$ and once by randomly initializing $\delta$ in the allowable perturbation ball. $k_1 = 5$, $k_2 = 20$ as described in A.1.

The MSD adversary used step sizes of $\alpha = (0.01, 0.2, 0.05)$ for the $(\ell_\infty, \ell_2, \ell_1)$ directions within a radius of $\epsilon = (0.3, 1.5, 12)$ for 100 iterations.

At test time, we increase the number of iterations to $(100, 200, 100)$ for $(\ell_\infty, \ell_2, \ell_1)$.

**CIFAR10**   The $\ell_\infty$ adversary used a step size $\alpha = 0.003$ within a radius of $\epsilon = 0.03$ for 40 iterations.

The $\ell_2$ adversary used a step size $\alpha = 0.05$ within a radius of $\epsilon = 0.5$ for 50 iterations.

The $\ell_1$ adversary used a step size $\alpha = 0.1$ within a radius of $\epsilon = 12$ for 50 iterations. $k_1 = 5$, $k_2 = 20$ as described in A.1.

The MSD adversary used step sizes of $\alpha = (0.003, 0.05, 0.05)$ for the $(\ell_\infty, \ell_2, \ell_1)$ directions within a radius of $\epsilon = (0.03, 0.3, 12)$ for 50 iterations. Note that the MSD model trained for $\ell_2$ radius of 0.3 is in fact robust to a higher radius of 0.5.

### B.2 TRAINING HYPERPARAMETERS

In this section, we describe the parameters used for adversarial training. For all the models, we used the SGD optimizer with momentum 0.9 and weight decay $5 \cdot 10^{-4}$.

**MNIST**   We train the models to a maximum of 20 epochs. We used a variation of the learning rate schedule from Smith (2018), which is piecewise linear from 0 to 0.1 over the first 7 epochs, down to 0.001 over the next 8 epochs, and finally back down to 0.0001 in the last 5 epochs.

**CIFAR10**   We used a variation of the learning rate schedule from Smith (2018) to achieve super-convergence in 50 epochs, which is piecewise linear from 0 to 0.1 over the first 20 epochs, down to 0.005 over the next 20 epochs, and finally back down to 0 in the last 10 epochs.

## C EXTENDED RESULTS

Here, we show the full tables which break down the overall adversarial error rates over individual attacks for both MNIST and CIFAR10, along with robustness curves for all models in the paper.

Table 3: Summary of adversarial accuracy results for MNIST

| | $P_\infty$ | $P_2$ | $P_1$ | B-ABS | ABS | Worst PGD | PGD Aug | MSD |
|---|---|---|---|---|---|---|---|---|
| Clean Accuracy | 99.1% | 99.4% | 98.9% | 99% | 99% | 98.9% | 99.1% | 98.2% |
| PGD-$\ell_\infty$ | 90.3% | 0.4% | 0.0% | - | - | 68.4% | 83.7% | 63.7% |
| FGSM | 94.9% | 68.6% | 6.4% | 85% | 34% | 82.4% | 90.9% | 81.8% |
| PGD-Foolbox | 92.1% | 8.5% | 0.1% | 86% | 13% | 72.1% | 85.7% | 67.9% |
| MIM | 92.3% | 14.5% | 0.1% | 85% | 17% | 73.9% | 87.3% | 71.0% |
| $\ell_\infty$ attacks ($\epsilon = 0.3$) | **90.3%** | 0.4% | 0.0% | 77% | 8% | 68.4% | 83.7% | 63.7% |
| PGD-$\ell_2$ | 83.8% | 87.0% | 70.8% | - | - | 85.3% | 87.9% | 84.2% |
| PGD-Foolbox | 93.4% | 89.7% | 74.4% | 63% | 87% | 86.9% | 91.5% | 86.9% |
| Gaussian Noise | 98.9% | 99.6% | 98.0% | 89% | 98% | 97.4% | 99.0% | 97.8% |
| Boundary Attack | 52.6% | 92.1% | 83.0% | 91% | 83% | 86.9% | 79.1% | 88.6% |
| DeepFool | 95.1% | 92.2% | 76.5% | 41% | 83% | 87.9% | 93.5% | 87.9% |
| Pointwise Attack | 74.3% | 97.4% | 96.6% | 87% | 94% | 92.7% | 89.0% | 95.1% |
| DDN | 82.7% | 87.0% | 70.8% | - | - | 85.1% | 85.2% | 84.3% |
| CWL2 | 88.2% | 88.1% | 75.5% | - | - | 85.2% | 87.5% | 85.1% |
| $\ell_2$ attacks ($\epsilon = 1.5$) | 45.3% | **87.0%** | 70.3% | 39% | 80% | 82.1% | 75.0% | 82.6% |
| PGD-$\ell_1$ | 51.8% | 49.9% | 71.8% | - | - | 66.5% | 57.4% | 64.8% |
| Salt & Pepper | 55.5% | 96.3% | 95.6% | 96% | 95% | 86.4% | 71.9% | 92.2% |
| Pointwise Attack | 2.4% | 66.4% | 85.2% | 82% | 78% | 60.1% | 17.1% | 72.8% |
| $\ell_1$ attacks ($\epsilon = 12$) | 1.4% | 43.4% | 71.8% | **82%** | 78% | 54.6% | 15.6% | 62.3% |
| All attacks | 1.4% | 0.4% | 0.0% | 39% | 8% | 53.7% | 15.6% | **58.7%** |

## C.1 MNIST RESULTS

**Expanded table of results** Table 3 contains the full table of results for all attacks on all models on the MNIST dataset. All attacks were run on a subset of the first 1000 test examples with 10 random restarts, with the exception of Boundary Attack, which by default makes 25 trials per iteration, and DDN attack, which does not benefit from restarts owing to a deterministic starting point. Note that the results for B-ABS and ABS models are from Schott et al. (2019), which uses gradient estimation techniques whenever a gradient is needed, and the robustness against all attacks for B-ABS and ABS is an upper bound based on the reported results. Further, these models are not evaluated with restarts, pushing the reported results even higher than actual.

## C.2 CIFAR10 RESULTS

**Expanded table of results** Table 4 contains the full table of results for all attacks on all models on the CIFAR10 dataset. All attacks were run on a subset of the first 1000 test examples with 10 random restarts, with the exception of Boundary Attack, which by default makes 25 trials per iteration, and DDN attack, which does not benefit from restarts owing to a deterministic starting point. Further note that salt & pepper and pointwise attacks in the $\ell_1$ section are technically $\ell_0$ attacks, but produce perturbations in the $\ell_1$ ball. Finally, it is clear here that while the training against an $\ell_1$ PGD adversary defends against said PGD adversary, it does not seem to transfer to robustness against other attacks.

## D COMPARISON WITH CONCURRENT WORK

In this section we compare the results of our trained MSD model with that of Tramèr & Boneh (2019), who study the theoretical and empirical trade-offs of adversarial robustness in various settings when defending against multiple adversaries. Training methods presented by them in their comparisons, namely $Adv_{avg}$ and $Adv_{max}$ closely resemble the naive approaches discussed in this paper: PGD-Aug and Worst-PGD respectively. We use the results as is from their work, and additionally compare the position of our MSD models at the revised thresholds used by Tramèr & Boneh (2019) without specially retraining them.

The results of Tables 5 and 6 show that the relative advantage of MSD over naive techniques does hold up. While we do make a comparison to the most relevant concurrent work for completeness, the

Table 4: Summary of adversarial accuracy results for CIFAR10

| | $P_\infty$ | $P_2$ | $P_1$ | Worst-PGD | PGD-Aug | MSD |
|---|---|---|---|---|---|---|
| Clean accuracy | 83.3% | 90.2% | 73.3% | 81.0% | 84.6% | 81.7% |
| PGD-$\ell_\infty$ | 50.3% | 48.4% | 29.8% | 44.9% | 42.8% | 49.8% |
| FGSM | 57.4% | 43.4% | 12.7% | 54.9% | 51.9% | 55.0% |
| PGD-Foolbox | 52.3% | 28.5% | 0.6% | 48.9% | 44.6% | 49.8% |
| MIM | 52.7% | 30.4% | 0.7% | 49.9% | 46.1% | 50.6% |
| $\ell_\infty$ attacks ($\epsilon = 0.03$) | 50.7% | 28.3% | 0.2% | 44.9% | 42.5% | 47.6% |
| PGD-$\ell_2$ | 59.0% | 62.1% | 28.9% | 64.1% | 66.9% | 66.0% |
| PGD-Foolbox | 61.6% | 64.1% | 4.9% | 65.0% | 68.0% | 66.4% |
| Gaussian Noise | 82.2% | 89.8% | 62.3% | 81.3% | 84.3% | 81.8% |
| Boundary Attack | 65.5% | 67.9% | 2.3% | 64.4% | 69.2% | 67.9% |
| DeepFool | 62.2% | 67.3% | 0.9% | 64.4% | 67.4% | 65.7% |
| Pointwise Attack | 80.4% | 88.6% | 46.2% | 78.9% | 83.8% | 81.4% |
| DDN | 60.0% | 63.5% | 0.1% | 64.5% | 67.7% | 66.2% |
| CWL2 | 62.0% | 71.6% | 0.1% | 66.9% | 71.5% | 68.7% |
| $\ell_2$ attacks ($\epsilon = 0.05$) | 57.3% | 61.6% | 0.0% | 61.7% | 65.0% | 64.3% |
| PGD-$\ell_1$ | 16.5% | 49.2% | 69.1% | 39.5% | 54.0% | 53.4% |
| Salt & Pepper | 63.4% | 74.2% | 35.5% | 75.2% | 80.7% | 75.6% |
| Pointwise Attack | 49.6% | 62.4% | 8.4% | 63.3% | 77.0% | 72.8% |
| $\ell_1$ attacks ($\epsilon = 12$) | 16.0% | 46.6% | 7.9% | 39.4% | 54.0% | 53.4% |
| All attacks | 15.6% | 27.5% | 0.0% | 34.9% | 40.6% | **46.1**% |

Table 5: Comparison with contemporary work on MNIST (higher is better). Results for all models except MSD are taken as is from Tramèr & Boneh (2019)

| | Vanilla | $Adv_\infty$ | $Adv_1$ | $Adv_2$ | $Adv_{avg}$ | $Adv_{max}$ | **MSD** |
|---|---|---|---|---|---|---|---|
| Clean accuracy | 99.4% | 99.1% | 98.9% | 98.5% | 97.3% | 97.2% | 98.2% |
| $\ell_\infty$ attacks ($\epsilon = 0.3$) | 0.0% | 91.1% | 0.0% | 0.4% | 76.7% | 71.7% | 63.7% |
| $\ell_2$ attacks ($\epsilon = 2.0$) | 12.4% | 12.1% | 50.6% | 71.8% | 58.3% | 56.0% | 67.4% |
| $\ell_1$ attacks ($\epsilon = 10$) | 8.5% | 11.3% | 78.5% | 68.0% | 53.9% | 62.6% | 70.0% |
| All attacks | 0.0% | 6.8% | 0.0% | 0.4% | 49.9% | 52.4% | **60.9**% |

following differences can bias the robust accuracies reported for the MSD models to relatively lower than expected (and correspondingly, the robust accuracies reported for the other models are relatively higher than expected):

1. **Use of random restarts:** We observe in our experiments that using up to 10 restarts for all our attacks leads to a decrease in model accuracy from 5 to 10% across all models. Tramèr & Boneh do not mention restarting their attacks for these models and so the results for models apart from MSD in Tables 5, 6 could potentially be lowered with random restarts.

2. **Different training and testing thresholds:** The MSD model for the MNIST dataset was trained at $\epsilon = (0.3, 1.5, 12)$ for the $\ell_\infty, \ell_2, \ell_1$ perturbation balls respectively, while Tramèr & Boneh (2019) tested at $\epsilon = (0.3, 2.0, 10)$. This may lower the robust accuracy at these thresholds for the MSD model, since it was not trained for that particular threshold. Likewise, the MSD model for CIFAR10 was also trained at $\epsilon = (0.03, 0.05, 12)$ for the $\ell_\infty, \ell_2, \ell_1$ perturbation balls respectively, while Tramèr & Boneh (2019) tested at $\epsilon = (\frac{4}{255}, 0, \frac{2000}{255})$.

3. **Different perturbation models:** For the CIFAR10 results in Table 6, $Adv_{avg}$ & $Adv_{max}$ models are trained and tested only for $\ell_1$ and $\ell_\infty$ adversarial perturbations, whereas the MSD model is robust to the union of $\ell_1, \ell_2$ and $\ell_\infty$, achieving a much harder task.

4. **Larger Suite of Attacks Used:** The attacks used by Tramèr & Boneh are PGD, EAD (Chen et al., 2017) and Pointwise Attack (Schott et al., 2019) for $\ell_1$; PGD, C&W (Carlini & Wagner, 2017) and Boundary Attack (Brendel et al., 2017) for $\ell_2$; and PGD for $\ell_\infty$ adversaries. We use a more expansive suite of attacks as shown in Appendix C. Some of the attacks like DDN, which proved to be strong adversaries in most cases, were not considered

Table 6: Comparison with contemporary work on CIFAR10 (higher is better). Results for all models except MSD are taken as is from Tramèr & Boneh (2019)

|  | Vanilla | $Adv_\infty$ | $Adv_1$ | $Adv_{avg}$ | $Adv_{max}$ | **MSD** |
|---|---|---|---|---|---|---|
| Clean accuracy | 95.7% | 92.0% | 90.8% | 91.1% | 91.2% | 82.1% |
| $\ell_\infty$ attacks ($\epsilon = \frac{4}{255}$) | 0.0% | 71.0% | 53.4% | 64.1% | 65.7% | 65.6% |
| $\ell_1$ attacks ($\epsilon = \frac{2000}{255}$) | 0.0% | 16.4% | 66.2% | 60.8% | 62.5% | 62.0% |
| All attacks | 0.0% | 16.4% | 53.1% | 59.4% | 61.1% | **61.7%** |

Table 7: Performance on CIFAR-10-C

|  | Accuracy |
|---|---|
| Standard model | 66.0% |
| $P_\infty$ | 75.0% |
| $P_2$ | 82.7% |
| $P_1$ | 57.8% |
| Worst-PGD | 70.8% |
| PGD-Aug | 76.8% |
| MSD | 74.2% |

by Tramèr & Boneh (2019) and thus were only used to attack the MSD models in Tables 5 and 6.

# E  ATTACKS OUTSIDE THE THREAT MODEL

In this section, we present some additional experiments exploring the performance of our model on attacks which lie beyond the threat model. Note that there is no principled reason why we would believe this to be the case (as most adversarial defenses tend to not generalize beyond the threat model defended against), and this is presented for exploratory reasons.

**Common corruptions**  We measure the performance of all the models on CIFAR-10-C, which is a CIFAR10 benchmark which has had common corruptions applied to it (e.g. noise, blur, and compression). We report the results in Table 7. We find that that, apart from the $P_1$ model, the rest achieve some improved robustness against these common corruptions above the standard CIFAR10 model.

**Defending against $\ell_1$ and $\ell_\infty$ and evaluating on $\ell_2$**  We also briefly study what happens when one trains against $\ell_1$ and $\ell_\infty$ threat models, while evaluating against the $\ell_2$ adversary. Specifically, we take the MSD approach on MNIST and simply remove the $\ell_2$ adversary from the threat model. This results in a model which has its $\ell_1$ and $\ell_\infty$ robust performance against a PGD adversary drop by 1% and its $\ell_2$ robust performance against a PGD adversary (which it was not trained for) drops by 2% in comparison to the original MSD approach on all three threat models.

As a result, we empirically observe that including the $\ell_2$ threat model in this setting actually improved overall robustness against all three threat models. Unsurprisingly, the $\ell_2$ performance drops to some degree, but the model does not lose all of its robustness.

