# OpenReview forum: "Adversarial Robustness Against the Union of Multiple Perturbation Models"
_ICLR.cc/2020/Conference — Reject_

### Official Review · AnonReviewer2 · 2019-10-19
**Official Blind Review #2**

**Rating:** 6

**Review:**

Summary of the paper: The paper describes adversarial training aiming to build models that are robust to multiple adversarial attacks - with L_1, L_2 and L_inf norms.  The method is a based on adversarial training against a union of adversaries. That union is created by taking (projected) gradient steps like PGD (Kurakin 2017), but choosing the maximal loss over GD steps for L1, L2, L_inf at each step.

Strengths: The topic is trendy and interesting. The proposed algorithm is simple and easy to implement. The experimental results demonstrate improvement over several baselines.

Weaknesses:
-- I am missing a more systematic comparisons to baseline defenses in the experiments. Figures 2 and 3 should have shown the accuracy as a function of radius also for PGD-aug, PGD-worst, Schott et al. Also what about comparisons to the latest SoTA defenses, e.g. recent baselines from from
www.robust-ml.org/defenses/.

-- An implicit expectation from this paper is that it addresses the key issue of  "Defend against one attack but face a different attack". The paper could have done more to advance our understanding of this issue. Specifically:

The approach improves over baselines for the "all attacks" mode, but under-performs compared with PGDaug and PGDworst when attacked with a single norm (Tab 1).

While this is expected and probably cannot be avoided, it leaves the reader with an unclear conclusion about risk tradeoffs. It would have been useful to clarify the regime of mixtures of attacks where the various approaches are best. For instance, if one uses a of mix attack samples from the three norms, what mixtures would it be best to defend using MSD, wand what mixtures would it be best to use PGD-aug? or ABS?



**Experience Assessment:**

I have published one or two papers in this area.

**Review Assessment: Checking Correctness Of Derivations And Theory:**

N/A

**Review Assessment: Checking Correctness Of Experiments:**

I assessed the sensibility of the experiments.

**Review Assessment: Thoroughness In Paper Reading:**

I read the paper at least twice and used my best judgement in assessing the paper.

---

> ### Author Response · Authors · 2019-11-07
> **Response to Reviewer #2**
>
> Thank you for your review and the provided suggestions.
>
> On the comparison to baseline defenses:
> Combining the robustness curves is a great suggestion, thank you. We do in fact have the accuracies as a function of radius for PGD-aug and PGD-worst (we had put them in the appendix as Figures 4-7 for lack of space), but we can certainly combine them into a single plot for MNIST and CIFAR10.
>
> As for the baselines at robust-ml.org, since the setting we study is the union of multiple threat models, we focus on baselines which also study defending against multiple threat models. To our knowledge, the only baseline on robust-ml.org which does this is the ABS model by Schott et al., which we explicitly compare to in our paper.
>
> On comparing the performances against individual attacks and the corresponding risk tradeoffs:
> The main point is that while comparing individual threat models leads to an unclear conclusion about risk tradeoffs as you pointed out, the conclusion for the reader is quite clear when measuring performance in the “all attacks” mode. This is the metric that makes the most sense, since this is exactly the robust optimization objective being minimized by all the algorithms, and has a simple interpretation as measuring performance when failure in even a single threat model is unacceptable.
>
> If one wishes to instead defend against a different mixture of attacks, then it makes more sense to change the robust optimization objective to reflect the different mixture of attacks using MSD, rather than trying to obtain it ad-hoc with PGD-aug or PGD-worst using a different threat model. Please see our general comment here for a more detailed discussion on measuring and comparing performance: https://openreview.net/forum?id=rklMnyBtPB&noteId=rJgRBxZ-iB

---

> > ### Author Response · Authors · 2019-11-12
> > **Updated paper**
> >
> > As noted above, we have made the following changes to the paper to reflect the feedback you have provided:
> >
> > + We have added a substantial discussion on the different risk tradeoffs between threat models that the various algorithms obtain. In summary, the simpler generalizations result in unclear tradeoffs, while MSD consistently minimizes worst-case performance over the union. This was added to the last paragraph of Section 5.
> >
> > + We have updated Figures 2 and 3 to allow for the more systemic comparison to baseline defenses by merging them with the corresponding figures in the Appendix, as suggested.

---

### Official Review · AnonReviewer3 · 2019-10-23
**Official Blind Review #3**

**Rating:** 1

**Review:**

This paper adversarially trains models against l_p norms where p is of there different values. They then propose a method which does somewhat better than the obvious way of adversarially training against more than one l_p perturbation.
The motivation for the paper is limited, in that they suggest previous works have suggested adversarial training itself "overfits" to the given l_p norm. This isn't surprising that it works, since the straightforward baseline works. They make it seem surprising by suggesting that ABS suggested adversarial training is doomed and cannot provide robustness to l_1, l_2, l_\infty norms simultaneously. The other motivation is that this is a step toward studying an expanded threat model, but the authors have not demonstrated that the learned representations are any bit more robust to common corruptions (could the authors show the generalization performance on CIFAR-10-C or generalization to unforeseen corruptions?). Without further evidence, we are left to believe this only helps for this narrow threat model. Overall the paper is deficient in creativity and generality, so I vote for rejection.

Small comments:

> take more time than a single norm,  it is a step closer towards the end goal of truly robust models, with adversarial robustness against all perturbations.
Please show model performance on CIFAR-10-C since if the model is more robust, it should hopefully be more robust to stochastic adversaries.

> has claimed that adversarial training “overfits” to the particular type of perturbation used to generate the adversarial examples
Wouldn't this be that l_\infty training fits specifically to l_\infty examples, not that robust optimization cannot handle more than one norm at a time? Who is claiming that?

> First, we show that even simple aggregations of different adversarial attacks can achieve competitive universal robustness against multiple perturbations models without resorting to complex architectures.
I am not sure this was in doubt. The phrase "universal robustness" is misleading.

How were the budgets chosen for l_2 and l_1? Those values seem small.

**Experience Assessment:**

I have published one or two papers in this area.

**Review Assessment: Checking Correctness Of Derivations And Theory:**

I assessed the sensibility of the derivations and theory.

**Review Assessment: Checking Correctness Of Experiments:**

I carefully checked the experiments.

**Review Assessment: Thoroughness In Paper Reading:**

I read the paper at least twice and used my best judgement in assessing the paper.

---

> ### Author Response · Authors · 2019-11-07
> **Response to Reviewer #3**
>
> Thank you for your feedback.
>
> Clarification on “overfitting”:
> First, we would like to clarify that we took the language of “adversarial training overfits to the L-infinity norm” directly from “Towards the first adversarially robust neural network model on MNIST” by Schott et al., which was published at last year’s ICLR, and is where the claim comes from (you can see this in the abstract). Of course, this is by no means a central point of the paper, and we merely wished to contextualize the result with relevant research on the same topic. We are quite willing to adjust the wording (e.g. the referenced phrasing with respect to overfitting and universal robustness).
>
> On the motivation and significance of MSD:
> While it is correct that the straightforward baselines work, they only work to some degree and are suboptimal when measuring their performance *with respect to the robust performance metric at which they attempt to minimize*, namely the robust optimization objective which is the performance against the union of threat models. On both MNIST and CIFAR10, we see a substantial increase in robust performance (5% and 6% respectively) on the union threat model from MSD over the baselines. This shows that the baselines, while they work to some extent, make various implicit tradeoffs that don’t actually minimize the robust objective that they are trying to minimize, and so MSD is a more direct, explicit way of minimizing the robust loss over the union adversary. The baselines themselves are also not consistent across the datasets: PGD-Aug performs poorly on MNIST, while PGD-Worst performs poorly on CIFAR10, whereas MSD is consistent across both problems.
>
> It is unfortunate that you think the approach is deficient in creativity and generality. Rather, we believe that the simplicity of the method adds to its strength, showing that even simple approaches can perform quite well without resorting to complex procedures. MSD is also general in that it can utilize any first-order iterative method for adversarial generation, and is not an image-specific defense (it is at least as generally applicable as the standard adversarial training approach for a single threat model).
>
> On generalizing to unforeseen corruptions:
> Defending against attacks outside of the threat model has never been a goal of adversarial training, and has little theoretical justification for why this would be the case. As such, performance comparisons on out-of-threat-model attacks like CIFAR-10-C, while potentially interesting, are completely orthogonal to the point of the paper. See our general comment here for a more detailed discussion: https://openreview.net/forum?id=rklMnyBtPB&noteId=rJgRBxZ-iB
>
> However, despite this, since CIFAR-10-C isn’t too expensive to evaluate, we ran this anyways just to see what happens and got the following mean accuracies:
>
> Standard model: 66.52%
> PGD-Worst: 70.8%
> PGD-Aug: 76.84%
> MSD: 74.22%
>
> So indeed, all of the approaches appear to improve model performance on CIFAR-10-C in comparison to standard training to some degree. However, because none of the models were explicitly trained to minimize these sorts of corruptions, we refrain from making any further conclusions.
>
> On the budgets chosen for L1 and L2:
> The chosen budgets all come from the literature. For MNIST, we chose the same budget as that used in “Towards the First Adversarially Robust Neural Network Model on MNIST” [Schott et al. 2019] in order to be directly comparable and most fair in the comparison. For CIFAR10 budgets comes from “Towards Evaluating the Robustness of Neural Networks” [Carlini & Wagner 2017], though we used a smaller L1 budget to account for the difference from L0 to L1 and to not entirely subsume the other threat models.

---

> > ### Author Response · Authors · 2019-11-12
> > **Updated paper**
> >
> > As noted above, we have made the following changes to the paper to reflect the feedback you have provided:
> >
> > + We have added the experiment requested, where we show model performance on CIFAR-10-C. This is in Appendix E, and reflects our previous comment.
> >
> > + We have adjusted the text to make it clear that Schott et al. use the fact that the L-infinity defense overfits to L-infinity perturbations as motivation for their paper, to avoid the misunderstanding that you brought up.

---

### Official Review · AnonReviewer1 · 2019-10-28
**Official Blind Review #1**

**Rating:** 3

**Review:**

The paper proposes to do adversarial training on multiple L_p norm perturbation models simultaneously, to make the model robust against various types of attacks.

[Novelty] I feel this is just a natural extension of adversarial training. If we define the perturbation set in PGD to be S, then in general S can be union of perturbation set of several L_p norm, and the resulting algorithm will be MSD (everytime you do a gradient update and then find the worst case projection in S). It would be interesting to study the convergence of this kind of algorithms, since S is no longer convex, the projection is trickier to define. Unfortunately this is not discussed in the paper.

In terms of experiments, this is an interesting data point to show that we can have a model that is (weakly) robust to L1, L2 and Linf norms simultaneously. However, the results are not surprising since there's more than 10% performance decreases compared to the original adversarial training under each particular attack. So it's still not clear whether we can get a model that simultaneously achieves L1, L2, Linf robust error comparable to original PGD training.

[Performance]
- It seems MSD is not always better than others (worst PGD and PGD Aug). For MNIST, MSD performs poorly on Linf norm and it's not clear why.
- There's significant performance drop in clean accuracy, especially MSD on MNIST data.

[Suggestions]
- As mentioned before, studying the convergence properties of the proposed methods will be interesting.
- It will be interesting if you can train on a set of perturbation models and make it also robust to another perturbation not in the training phase. For instance, can we apply the proposed method to L{1,inf} in training and generalize to L2 perturbation?

=====
Thanks for the response. I still have concerns about novelty so would like to keep my rating unchanged.



**Experience Assessment:**

I have published in this field for several years.

**Review Assessment: Checking Correctness Of Derivations And Theory:**

I carefully checked the derivations and theory.

**Review Assessment: Checking Correctness Of Experiments:**

I carefully checked the experiments.

**Review Assessment: Thoroughness In Paper Reading:**

I read the paper thoroughly.

---

> ### Author Response · Authors · 2019-11-07
> **Response to Reviewer #1**
>
> Thank you for your feedback. We definitely aimed for this to be a “natural” extension of adversarial, and we view the simplicity of the approach to be an advantage of the approach over relying on more complex methods. As a minor note, the extension goes beyond finding the worst-case projection: it is important to also consider the individual steepest descent directions for each threat model, so there is no singular gradient step like in PGD.
>
> On convergence:
> We do agree that studying the convergence properties could be interesting, however, this is not our focus and is out of the scope of this paper. This is actually a fairly complex problem: the convergence properties of steepest descent for a *single* norm (to our knowledge) for deep networks is not quite known.
>
> On performance:
> The main point is that while comparing individual threat models leads to an unclear conclusion about risk tradeoffs as you pointed out, the conclusion for the reader is clear when measuring performance in the “all attacks” mode. This is the metric that makes the most sense, since this is exactly the robust optimization objective being minimized by all the algorithms, and has a simple interpretation as measuring performance when a failure in even a single threat model is unacceptable. We will adjust the paper accordingly to make this more obvious. Please see our general comment here for a more detailed discussion on comparing performances, as well as on generalizing outside the threat model used during training: https://openreview.net/forum?id=rklMnyBtPB&noteId=rJgRBxZ-iB

---

> > ### Author Response · Authors · 2019-11-12
> > **Updated paper**
> >
> > As noted above, we have made the following changes to the paper to reflect the feedback you have provided:
> >
> > + We have added the experiment requested, where we train on Linfinity and L1, while evaluating on L2. This is in Appendix E, and reflects the additional discussion in the comment above.

---

### Public Comment · ~Anthony_Wittmer1 · 2019-09-29
**small question**

Great work.

For the combination of mutiple pertubation by PGD augmentation with all perturbations , the method seems like another type of ensemble adversarial training[1], which trains with different adversaries.

For multi steepest descent(MSD), how to control the adversarial perturbations in the various norm bounded? At each iteration, the norm to be chosen may be different, and later norm may affect the adversarial perturbations generated by previous norm.

[1] Ensemble Adversarial Training: Attacks and Defenses. ICLR 2018

---

> ### Author Response · Authors · 2019-09-30
> **MSD : adversarial perturbation w.r.t. iteration**
>
> Thank you for your interest. At each iteration, MSD aims to maximize the loss of adversarial perturbation that is generated after taking a step in the direction of either P_1, P_2 or P_inf adversary and projecting back to the corresponding perturbation ball. The decision of the next iteration is agnostic of what happened in the previous one, so any decision taken 'later' is only taken to improve open the 'previous' loss value. In practice, MSD is found to benefit by 'switching' norm decisions during the descent iterations.

---

> > ### Public Comment · ~Anthony_Wittmer1 · 2019-09-30
> > **Clipping operation**
> >
> > Sorry, it may not be clear.
> >
> > In order to control the adversarial perturbations in the specific norm bounded, it needs the clipping operation at the ending of attack. Later clipping operation may affect the adversarial perturbations generated by previous norm.
> >
> > For example, previous norm is L_2 norm, so on some pixels, the perturbations  are  zero, and on some pixels, the perturbations  are  larger than 10. If the next norm is L_{\infty } with the epsilon 8, the clipping operation will reduce the perturbations (larger than 10) of the previous attack to 8.

---

> > > ### Author Response · Authors · 2019-10-08
> > > **Clarification**
> > >
> > > Yes, you are correct. More precisely, the algorithm does a projection back to the relevant l_p ball (which turns out to be clipping in case of l_inf). However, as mentioned above, the "switch in l_p choice" happens only according to the "loss value" of the current step. So, even though it may seem to you that the perturbation value is getting 'reduced', the iterant is actually moving to a point with a higher loss value, and such transitions are found to be beneficial to the training.

---

> > > > ### Public Comment · ~Anthony_Wittmer1 · 2019-10-08
> > > > **Thanks for the reply.**
> > > >
> > > > Thanks for the reply.
> > > >
> > > > It looks like a greedy algorithm, which chooses the worst case for various (P_1, P_2 or P_inf) norm at each iteration. However, in this way, the adversarial example at the last iteration is not necessarily the worst case for the attack process. Maybe reinforcement learning can help.
> > > >
> > > > Do the authors have any insight about integating different norm to one specific norm, which may take less traning time rather than increasing the training time by the number of various norm?

---

> > > > > ### Author Response · Authors · 2019-10-21
> > > > > **Multiple Norms**
> > > > >
> > > > > Defending against one norm, as you probably already know, only defends against a single adversary. While defending against multiple norms does, in fact, take more time than a single norm,  it is a step closer towards the end goal of truly robust models, with adversarial robustness against all perturbations.

---

> > > > > > ### Public Comment · ~Anthony_Wittmer1 · 2019-10-21
> > > > > > **Yes**
> > > > > >
> > > > > > Yes, it is effective to improve the robustness by taking mutiple time  to train with mutiple adversaries.
> > > > > >
> > > > > > However, it is better to develop well-generalized model to align better with **human perception**, rather than roughly taking plenty of time  to train with all adversaries. Besides the L_p norm restricted adversaries, there are some unrestricted adversaries, such as spatial attack, semantic attack and so on.

---

### Author Response · Authors · 2019-11-07
**General Response to R1, R2, R3**

A common theme throughout the reviews focuses on the performance of the MSD trained model on different threat models. Namely, 1) the performance of the model on individual threat models (subsets of the considered threat region), and 2) the ability of the model to generalize beyond the threat model it was trained on. We discuss these points in this comment, but at a high level, the main message is that the most natural metric to use for evaluating the various algorithms is the *robust objective being minimized*, which MSD does best. All other metrics (individual threat models or attacks outside the threat model) are simply not what any of these algorithms are trying to minimize. We will adjust the text to make this clear.

1) On the performance for individual threat models:
As you likely are well aware, there is very rarely free lunch in adversarial robustness: at some point, tradeoffs between various metrics (e.g. standard vs robust accuracy, or robust performance against different threat models) become inevitable. As pointed out, the other training procedures (Worst PGD and PGD Aug) do achieve different trade-offs between the various threat models, and for specific individual threat models, can achieve better performance on those specific threat models.

However, the tradeoffs that these methods achieve are suboptimal when measuring performance against the *union* of multiple threat models, which is the goal of this paper (and importantly, also the mathematical objective for the robust optimization problem). By no means do we claim MSD to have, for example, the best performance on L-infinity robustness for MNIST. An adversarial attack on the union of threat models is successful if it succeeds within *any* of the threat models. This is the objective that all the methods (MSD, Worst PGD, PGD Aug) attempt to minimize, and this is where we see the advantage of MSD: it is able to achieve the best performance when the union of threat models is taken as a whole.

The takeaway here is that yes, there are indeed different tradeoffs obtained by the various methods, however, MSD is most effective at finding the tradeoff that maximizes the goal of robust performance to the union of perturbation sets, which is directly the robust optimization objective and thus the most natural metric. On the other hand, the alternatives (Worst PGD and PGD Aug) find some suboptimal tradeoff that doesn’t quite maximize the robust optimization objective for the union of multiple sets, despite being seemingly obvious ways to do so.

We do not claim to achieve top performance on individual threat models or standard accuracy (neither of which is directly the goal of the robust optimization problem for the union threat model), and so while it would be nice if this were the case, it is certainly not expected and may not even be possible. To give an analogous example, when studying threat models for a single norm, we do not carve up the threat model into subsets and compare performance within the various subsets (at least, not beyond plotting robustness curves for different radii, which we do in this paper).

If it is still insisted that we compare the individual threat models: as Reviewer 2 discussed, it becomes unclear how to evaluate the various tradeoffs. On the other hand, the robust optimization objective, or the performance against the union of threat models, is directly the goal of all these algorithms and leaves the reader with a clear interpretation: it measures the performance of the model when a failure under any threat model constitutes an overall failure, which MSD is able to do best.

2) On the ability of the defense to generalize beyond the threat model:
Being able to generalize beyond the threat model on which a model has been trained has never been a goal of adversarial training. In most cases, there is little to no principled reason for why we would believe this to occur, and empirically the answer in the adversarial examples literature tends to be that it does not. Generalizing beyond the threat model used in training is completely orthogonal and not at all a goal of this paper, let alone a goal of adversarial training.

Rather, the goal of this paper is to present a structured way in which the threat model can be expanded *during* training, as this is the only scenario in which we would expect the defense to generalize. The way to defend against a new threat model would be to add it to the set of threat models and use one of the methods presented in this paper to defend against the union of threat models.

---

### Author Response · Authors · 2019-11-12
**Revisions to the paper**

In light of the reviewer feedback, we have made a number of changes to the paper which we outline in this comment. These changes reflect nearly all of the constructive suggestions that we have received from the reviewers.

Of course, we are aware that there seems to be a fundamental disagreement over the importance of evaluating an adversarial defense outside of its threat model, which we discussed in a longer, earlier comment. Despite this being a completely non-standard metric for evaluating adversarial defenses throughout the literature, we have gone ahead and incorporated all of the suggestions for adding these experiments into the updated paper. Note that all of the relevant work that we compare to does *not* do any evaluation of this sort, so it is rather unprecedented for this to suddenly become a necessary requirement.


Summary of changes:

+ We have added a substantial discussion on the different risk tradeoffs between threat models that the various algorithms obtain, as requested by Reviewer 2. In summary, the simpler generalizations result in unclear tradeoffs, while MSD consistently minimizes worst-case performance over the union. This was added to the last paragraph of Section 5.

+ We have updated Figures 2 and 3 to allow for the more systemic comparison to baseline defenses by merging them with the corresponding figures in the Appendix, as requested by Reviewer 2.

+ We have added the experiment requested by Reviewer 3, where we show model performance on CIFAR-10-C. This is in Appendix E, and reflects the additional discussion we've had with Reviewer 3 on OpenReview.

+ We have adjusted the text to make it clear that Schott et al. use the fact that the L-infinity defense overfits to L-infinity perturbations as motivation for their paper, to avoid the misunderstanding that Reviewer 3 brought up.

+ We have added the experiment requested by Reviewer 1, where we train on Linfinity and L1, while evaluating on L2. This is in Appendix E, and reflects the additional discussion we've had with Reviewer 1 on OpenReview.

Lastly, we remind our reviewers that we have already stuck to a very high standard for an extensive adversarial evaluation within the threat model, following best practices in the field and using a wide variety of gradient and non-gradient based attacks, which is among the most comprehensive evaluations present in the literature.

---

### Decision · Program_Chairs · 2019-12-19

**Decision:**

Reject

**Comment:**

Thanks to the authors for submitting the paper and providing further explanations and experiments. This paper aims to ensure robustness against several perturbation models simultaneously. While the authors' response has addressed several issues raised by the reviewers, the concern on the lack of novelty remains. Overall, there is not enough support among the reviewers for the paper to be accepted.